# Changes in the top 25 reasons for primary care visits during the COVID-19 pandemic in a high-COVID region of Canada

Ellen Stephenson[1]*, Debra A. Butt[1,2], Jessica Gronsbell[1,3], Catherine Ji[1,4], Braden O'Neill[1,5,6], Noah Crampton[1,4], Karen Tu[1,4,7]

**1** Department of Family and Community Medicine, University of Toronto, Toronto, Ontario, Canada, **2** Scarborough Health Network, Toronto, Ontario, Canada, **3** Department of Statistical Sciences, University of Toronto, Ontario, Canada, **4** Toronto Western Hospital Family Health Team, University Health Network, Toronto, Ontario, Canada, **5** Department of Family and Community Medicine, St. Michael's Hospital, Toronto, Ontario, Canada, **6** MAP Centre for Urban Health Solutions, Li Ka Shing Knowledge Institute, St. Michael's Hospital, Toronto, Ontario, Canada, **7** North York General Hospital, Toronto, Ontario, Canada

\* ellen.stephenson@utoronto.ca

**Data Availability Statement:** The research ethics approval for the use of UTOPIAN data does not permit making the data publicly available, as the data contain potentially identifiable patient

## Abstract

### Purpose

We aimed to determine the degree to which reasons for primary care visits changed during the COVID-19 pandemic.

### Methods

We used data from the University of Toronto Practice Based Research Network (UTOPIAN) to compare the most common reasons for primary care visits before and after the onset of the COVID-19 pandemic, focusing on the number of visits and the number of patients seen for each of the 25 most common diagnostic codes. The proportion of visits involving virtual care was assessed as a secondary outcome.

### Results

UTOPIAN family physicians ($N$ = 379) conducted 702,093 visits, involving 264,942 patients between March 14 and December 31, 2019 (pre-pandemic period), and 667,612 visits, involving 218,335 patients between March 14 and December 31, 2020 (pandemic period). Anxiety was the most common reason for visit, accounting for 9.2% of the total visit volume during the pandemic compared to 6.5% the year before. Diabetes and hypertension remained among the top 5 reasons for visit during the pandemic, but there were 23.7% and 26.2% fewer visits and 19.5% and 28.8% fewer individual patients accessing care for diabetes and hypertension, respectively. Preventive care visits were substantially reduced, with 89.0% fewer periodic health exams and 16.2% fewer well-baby visits. During the pandemic, virtual care became the dominant care format (77.5% virtual visits). Visits for anxiety and depression were the most common reasons for a virtual visit (90.6% virtual visits).

information. Researchers interested in accessing EMR data from the UTOPIAN Data Safe Haven for research can apply to do so at: https://www.dfcm. utoronto.ca/getting-utopian-support.

**Funding:** This study was made possible by a CIHR Operating Grant: COVID-19 Mental Health & Substance Use – Matching Access to Service with Needs: grant #450302 (PI: KT). The funders had no role in study design, data collection and analysis, decision to publish, or preparation of the manuscript.

**Competing interests:** The authors have declared that no competing interests exist.

## Conclusion

The decrease in primary care visit volumes during the COVID-19 pandemic varied based on the reason for the visit, with increases in visits for anxiety and decreases for preventive care and visits for chronic diseases. Implications of increased demands for mental health services and gaps in preventive care and chronic disease management may require focused efforts in primary care.

## Introduction

The COVID-19 pandemic has presented unprecedented challenges in primary health care worldwide since it was first declared in March 2020 [1–5]. Governments around the world have implemented policies to prioritize the use of health care resources to treat patients with COVID-19 and to prevent the spread of the disease [6], including decreasing non-COVID hospital admissions [7] and delaying elective surgeries [8]. In primary care settings, physicians were encouraged to triage medical appointments, prioritize services that would prevent acute care hospitalization, and increase their use of virtual care [1, 9–12]. Although changes in primary care practice occurred rapidly in response to the COVID-19 pandemic, the implications of these changes have yet to be determined.

The World Organization of Family Doctors (WONCA) Europe has identified a lack of evidence regarding the management of non-COVID-19 patients in primary care during the pandemic [13]. To fill this gap, we first need to know the reasons why patients seek primary care, and how this may have changed during the COVID-19 pandemic. A 2018 review found that acute respiratory infections, chronic diseases (i.e., hypertension, diabetes, arthritis), anxiety or depression, routine health maintenance, and back pain were among the most common clinician reported reasons for a primary care visit [14]. In Canada, over half of family physician visits involve patients with chronic diseases, and this increases with advancing age [15, 16]. There is some evidence that older Canadians accessed more primary care visits during the pandemic than younger Canadians [17, 18], however the reasons for those visits are unknown. Without knowing the degree to which different types of primary care services were impacted by the pandemic, it is difficult to estimate the value of the forgone care and the potential implications for health outcomes.

The goal of the current study was to determine the impact of COVID-19 on the most common reasons for primary care visits in Ontario, Canada's largest provincial health system. We examined a region of Ontario with the greatest number of COVID-19 cases and the extent to which the effect of the COVID-19 pandemic and the uptake of virtual care varied based on the reason for the primary care visit.

## Methods

### Study design

We used a repeated cross-sectional design in which primary care visits for a fixed cohort of family physicians were sampled after the onset of the COVID-19 pandemic (March 14-December 31, 2020) and the same time the year before (March 14-December 31, 2019). The same time period was used in 2019 and 2020 to account for any potential seasonal variation in reasons for primary care visits [19]. The start of the pandemic period was defined as March 14, 2020 when the Ontario Ministry of Health introduced new physician billing codes for the

provision of virtual care via telephone or video [20]. Prior to this, virtual visits in primary care were not widely utilized [21].

## Data source and setting

We used data from the University of Toronto Practice-Based Research Network (UTOPIAN) Data Safe Haven, a primary care electronic medical record (EMR) database [22]. This database includes records from family medicine clinics in Ontario, Canada, with most physicians practicing in the Greater Toronto Area, a region of Ontario that has consistently had the highest number of COVID-19 cases [23]. The number of confirmed cases of COVID-19 per capita was more than 1.5 times higher for public health units in Peel (2.5 per million) and Toronto (1.9 per million) relative to the provincial average (1.2 per million) as of December 31, 2020 [24]. Public health measures implemented to control the spread of COVID-19 included closures of schools and non-essential businesses, travel restrictions, instructions to only leave home for essential purposes such as accessing medical services, mandatory mask wearing in indoor public spaces, and limiting the size of social gatherings.

Ontario has a government run single payer health insurance system that provides coverage for hospital and primary care services to most residents through the Ontario Health Insurance Plan (OHIP). Family physicians use their EMR system to bill OHIP for the services they provide, such that an OHIP billing code and corresponding diagnostic code are recorded for all encounters documented in the EMR database. OHIP service codes for diabetes, prenatal care and periodic health exams do not require an accompanying diagnostic code [25]. In these instances, we inferred the relevant diagnostic code if none was billed along with one of these OHIP service codes (S1 Appendix).

A cohort of physicians and patients who met minimum data quality requirements and were active within the EMR during the study period were selected for inclusion in the current study (Fig 1). To be eligible to contribute EMR data, family physicians had to meet minimum standards for data quality (S2 Appendix) and have started using their EMR prior to March 14,

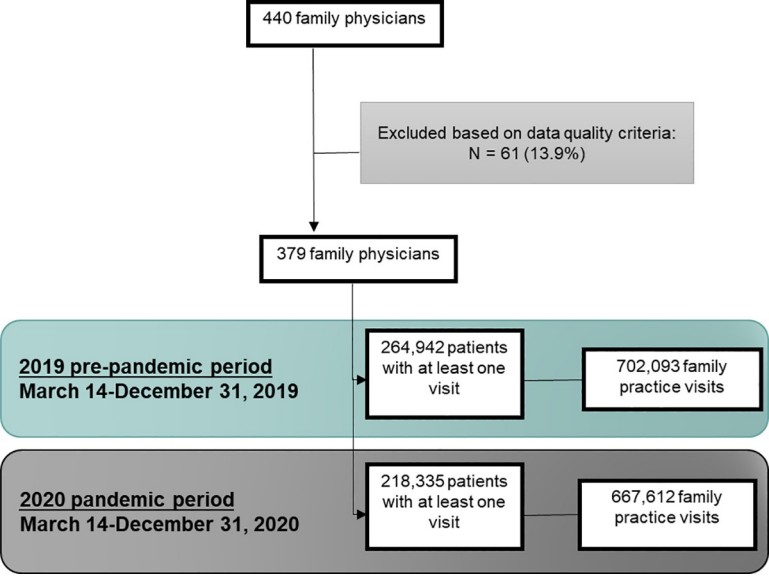

**Fig 1. Record selection process.**

2019. All patients who were registered to an eligible physician, had a valid age and sex recorded in the EMR, and had at least one visit during the observation period were eligible for inclusion. Billing records with an OHIP service code for an office or virtual visit were included (S3 Appendix).

## Outcome measures

We selected the 25 most frequently used diagnostic codes from March 14-December 31, 2019 (pre-pandemic period) and compared these to the 25 most frequently used diagnostic codes for the same time period in 2020 (pandemic period). To understand the impact of COVID-19 on the use of each of the top 25 diagnostic codes, we compared three outcomes during the pre-pandemic and pandemic periods: (i) the number of times each diagnostic code was billed (number of visits), (ii) the number of unique patients seen at least once for a visit associated with each diagnostic code (number of visitors), and (iii) the number of visits relative to the number of patients who visited at least once (visits per visitor). Separating the change in the number of visits into a change in the number of unique visitors and a change in the number of visits per visitor allowed us to examine pandemic-related changes in healthcare use (vs. non-use) separately from the effects of the pandemic on intensity of healthcare use. This is an important distinction to make because healthcare use and intensity have been found to be affected by different factors [26].

To better understand not only changes in visit volume, but also changes in visit format that occurred during the pandemic, we examined the uptake of virtual care by reason for visit. Because virtual care was not widely available before the pandemic, we calculated the proportion of visits that occurred via telephone or video during the pandemic period for each diagnosis code as a secondary outcome measure.

## Statistical analysis

We examined each outcome in the pre-pandemic and pandemic periods along with the percentage change from the pre-pandemic to pandemic period. To further understand the impact of the pandemic on healthcare volume at the physician-level, we fit Poisson regression models to estimate the relative difference in (i) the number of visits and (ii) the number of visitors per physician in the pre-pandemic and pandemic periods. To account for within-physician correlation over time, we used generalized estimating equations (GEEs) to estimate the model parameters [27, 28] and adopted a working exchangeable correlation structure. We calculated standard errors, two-sided p-values, 95% confidence intervals (CIs) for the rate ratios using the robust sandwich estimator. Statistical significance was based on two-sided p-values at the 0.05 level.

## Ethics approval

This study was approved through the University of Toronto (#40129) and North York General Hospital (#20–0044) research ethics boards. Physicians in this study provided written informed consent to have their EMR data extracted, de-identified, and used for research purposes; patients can opt out of uses for research.

## Results

### Sample description

Sociodemographic characteristics of patients and physicians included in the current study are summarized in Table 1. All physicians practiced as part of a blended capitation model [29].

**Table 1. Physician and patient characteristics.**

| | UTOPIAN Family Physicians (N = 379) | | Ontario Family Physicians |
|---|---|---|---|
| | N | % | % |
| **Sex** | | | |
| Female | 232 | 61.2 | 49.5 |
| Male | 147 | 38.8 | 50.5 |
| **Age (years)** | | | |
| < 30 | <5 | <1 | † |
| 30–44 | 153 | 40.4 | 33.3 |
| 45–64 | 156 | 41.2 | 48.1 |
| 65–74 | 37 | 9.8 | 14.5 |
| ≥ 75 | <5 | <1 | 3.5 |
| Missing | 29 | 7.7 | † |
| **Medical School Graduation** | | | |
| Canada | 341 | 90.0 | 55.3 |
| Other (including United States) | 38 | 10.0 | 25.5 |

| | Patients with at least one visit during pre-pandemic period (N = 264,942) | | Patients with at least one visit during pandemic period (N = 218,335) | | Ontario population |
|---|---|---|---|---|---|
| | N | % | N | % | % |
| **Sex** | | | | | |
| Female | 154,706 | 58.4 | 130,984 | 60.0 | 50.6 |
| Male | 110,236 | 41.6 | 87,351 | 40.0 | 49.4 |
| **Age (years)** | | | | | |
| <19 | 39,000 | 14.7 | 28,229 | 12.9 | 21.5 |
| 19–34 | 43,603 | 16.5 | 37,769 | 17.3 | 21.2 |
| 34–49 | 52,182 | 19.7 | 43,834 | 20.1 | 19.4 |
| 50–64 | 62,758 | 23.7 | 52,020 | 23.8 | 20.6 |
| >64 | 67,399 | 25.4 | 56,483 | 25.9 | 17.3 |
| **Neighborhood income quintile** | | | | | |
| 1 (Lowest income) | 51,882 | 19.6 | 42,518 | 19.3 | |
| 2 | 45,325 | 17.1 | 37,070 | 16.9 | |
| 3 | 44,997 | 17.0 | 36,901 | 16.9 | |
| 4 | 47,894 | 18.0 | 39,292 | 18.1 | |
| 5 (Highest income) | 68,168 | 25.7 | 55,757 | 25.7 | |
| Missing | 6,676 | 2.5 | 6,797 | 3.1 | |

Ontario population estimates for age and sex are based on Statistics Canada data for July 1, 2019. Physician age, sex, and medical graduation for Ontario family physicians are based on results from a 2021 report using the ICES Physician Database [17]. † Not available in source used for the Ontario population.

Patients who were female, older age, and higher income were overrepresented in the sample relative to the Ontario population, but this is consistent with characteristics of health care users [30]. UTOPIAN family physicians were more likely to be younger, female, and Canadian medical graduates relative to family physicians across Ontario. Family physicians included in the current study (N = 379) conducted 702,093 primary care visits, involving 264,942 patients with at least one visit (mean number of visits per visitor = 2.65) during the pre-pandemic period. During the pandemic period, the total number of visits dropped to 667,612 (-4.9%) and the total number of patients with at least one visit dropped to 218,335 (-17.6%), such that the mean number of visits per visitor increased to 3.06.

## Most frequent reasons for visit

The 25 most frequently billed diagnostic codes associated with visits in the pre-pandemic period are listed in Table 2. The most common reasons for visit during the pandemic were similar but included the new diagnostic OHIP code for coronavirus, which was the 19th most frequent reason for a visit during the 2020 pandemic period (N = 7,523 visits; 1.1% of total visit volume). Anxiety, diabetes, and hypertension remained the most common reasons for visit both pre-pandemic and during the pandemic, whereas periodic health exam and common cold were the 4th and 5th most common reason pre-pandemic but dropped to 39th and 13th place during the pandemic (Table 2).

## Changes in the number of visits by reason for visit

Relative to the pre-pandemic period, the total number of visits during the pandemic period was lower for chronic conditions (diabetes, hypertension, osteoarthritis), preventive care

**Table 2. Frequency of the most common diagnostic codes billed for primary care visits.**

| Diagnostic code | Most common associated description* | Pre- COVID (March 14-December 31, 2019) | | COVID (March 14-December 31, 2020) | | Difference |
|---|---|---|---|---|---|---|
| | | Rank | Number of visits (% of total visit volume) | Rank | Number of visits (% of total visit volume) | Percentage change in number of visits |
| 300 | Anxiety | 1 | 45,629 (6.5) | 1 | 61,121 (9.2) | 34.0 |
| 250 | Diabetes | 2 | 43,548 (6.2) | 2 | 33,246 (5.0) | -23.7 |
| 401 | Hypertension | 3 | 40,414 (5.8) | 3 | 29,824 (4.5) | -26.2 |
| 917 | Periodic Health Exam | 4 | 31,739 (4.5) | 39 | 3,479 (0.5) | -89.0 |
| 460 | Common Cold | 5 | 23,010 (3.3) | 13 | 11,165 (1.7) | -51.5 |
| 787 | Abdominal Pain | 6 | 21,240 (3.0) | 4 | 24,384 (3.7) | 14.8 |
| 781 | Musculoskeletal Pain | 7 | 21,046 (3.0) | 5 | 21,703 (3.3) | 3.1 |
| 916 | Well Baby Care | 8 | 19,626 (2.8) | 8 | 16,451 (2.5) | -16.2 |
| 799 | Ill-defined Conditions | 9 | 19,387 (2.8) | 6 | 20,547 (3.1) | 6.0 |
| 650 | Pregnancy/Delivery | 10 | 17,986 (2.6) | 7 | 19,236 (2.9) | 6.9 |
| 691 | Eczema | 11 | 13,295 (1.9) | 9 | 13,824 (2.1) | 4.0 |
| 724 | Back Pain | 12 | 11,647 (1.7) | 11 | 11,735 (1.8) | 0.8 |
| 715 | Osteoarthritis | 13 | 10,287 (1.5) | 16 | 8,551 (1.3) | -16.9 |
| 785 | Chest Pain | 14 | 10,031 (1.4) | 10 | 11,950 (1.8) | 19.1 |
| 780 | Vertigo, Dizziness, Headache | 15 | 9,845 (1.4) | 12 | 11,544 (1.7) | 17.3 |
| 895 | Family Planning | 16 | 9,824 (1.4) | 15 | 9,216 (1.4) | -6.2 |
| 786 | Cough, Epistaxis | 17 | 9,553 (1.4) | 17 | 8,509 (1.3) | -10.9 |
| 896 | Immunization | 18 | 8,668 (1.2) | 26 | 5,846 (0.9) | -32.6 |
| 599 | Hematuria | 19 | 8,010 (1.1) | 14 | 10,481 (1.6) | 30.8 |
| 311 | Depression | 20 | 7,979 (1.1) | 18 | 8,491 (1.3) | 6.4 |
| 847 | Strain, Sprain, Back Pain | 21 | 6,990 (1.0) | 23 | 6,332 (0.9) | -9.4 |
| 727 | Tendonitis, Bunion | 22 | 6,979 (1.0) | 25 | 6,156 (0.9) | -11.8 |
| 796 | Fatigue | 23 | 6,357 (0.9) | 21 | 6,723 (1.0) | 5.8 |
| 709 | Mole, Other Disorders of Skin | 24 | 6,063 (0.9) | 22 | 6,433 (1.0) | 6.1 |
| 272 | Hypercholesterolemia | 25 | 5,743 (0.8) | 27 | 5,639 (0.8) | -1.8 |
| Top 25 | | < 26 | 41,4896 (59.1) | | 372,586 (55.8) | -10.6 |
| All | All visits | | 702,093 (100.0) | | 667,612 (100.0) | -4.9 |

These 25 diagnostic codes were selected based on how frequently they were billed in March 14-December 31, 2019. The top 25 diagnostic codes used in the 2020 period also included codes for coronavirus (N = 7,523; 1.1%), menstrual disorders (N = 6,825; 1.0%), and cystitis (N = 6,197; 0.9%).

*The most common description associated with each diagnostic code is provided above; a full list of all diagnoses associated with each code is provided in the supplementary appendix. One visit was counted per patient per date and less than 1% of visits involved the use of multiple diagnostic codes.

(periodic health exams, immunizations, well-baby care), common cold, and family planning, higher for anxiety, abdominal pain, chest pain, headache, and hematuria, and unchanged for pregnancy/delivery, eczema, musculoskeletal pain, back pain, fatigue, and other ill-defined conditions (Table 2). The largest change in visit volume was observed for periodic health exams (-89.0%), which accounted for 4.5% of all visits pre-pandemic, but ceased almost entirely after the onset of the pandemic (0.5% of visits).

## Changes in the number of patients accessing care by reason for visit

Although the number of visits for some conditions increased, this did not always mean that more individual patients were accessing care. Anxiety and hematuria were the only diagnostic codes with more individual patients accessing care during the pandemic period than in the pre-pandemic period. In most cases there was a decline in the number of individual patients accessing care (i.e., visiting at least once), and this was accompanied by an increase in the number of contacts for those who did visit their family physician (i.e., more visits per visitor; Table 3). Decreases in the number of patients with at least one visit during the pandemic were especially pronounced for periodic health exams (-89.1%), common cold (-55.6%), immunizations (-33.3%), hypertension (-28.8%), osteoarthritis (-22.9%), and diabetes (-19.5%). Diabetes

**Table 3. Number of visitors and visits per visitor by reason for visit.**

| Type of reason for visit | Diagnostic code | Most common associated description* | Number of visitors | | | Number of visits per visitor | | |
|---|---|---|---|---|---|---|---|---|
| | | | Pre COVID | COVID | % Change | Pre COVID | COVID | % Change |
| Mental health | 300 | Anxiety | 25962 | 28610 | 10.2 | 1.76 | 2.14 | 21.6 |
| | 311 | Depression | 4795 | 4645 | -3.1 | 1.66 | 1.83 | 10.2 |
| Chronic disease | 401 | Hypertension | 24922 | 17734 | -28.8 | 1.62 | 1.68 | 3.7 |
| | 250 | Diabetes | 21585 | 17380 | -19.5 | 2.02 | 1.91 | -5.4 |
| | 715 | Osteoarthritis | 7609 | 5867 | -22.9 | 1.35 | 1.46 | 8.1 |
| Preventive care | 917 | Periodic Health Exam | 31568 | 3437 | -89.1 | 1.01 | 1.01 | 0.0 |
| | 916 | Well Baby Care | 8093 | 6553 | -19.0 | 2.43 | 2.51 | 3.3 |
| | 896 | Immunization | 7418 | 4947 | -33.3 | 1.17 | 1.18 | 0.9 |
| Other | 460 | Common Cold | 19474 | 8650 | -55.6 | 1.18 | 1.29 | 9.3 |
| | 787 | Abdominal Pain | 16877 | 16509 | -2.2 | 1.26 | 1.48 | 17.5 |
| | 781 | Musculoskeletal Pain | 16240 | 14598 | -10.1 | 1.30 | 1.49 | 14.6 |
| | 799 | Ill-defined Conditions | 16007 | 15444 | -3.5 | 1.21 | 1.33 | 9.9 |
| | 650 | Pregnancy/Delivery | 4960 | 4952 | -0.2 | 3.63 | 3.88 | 6.9 |
| | 691 | Eczema | 11404 | 10754 | -5.7 | 1.17 | 1.29 | 10.3 |
| | 724 | Back Pain | 8672 | 7328 | -15.5 | 1.34 | 1.60 | 19.4 |
| | 785 | Chest Pain | 8413 | 8685 | 3.2 | 1.19 | 1.38 | 16.0 |
| | 786 | Cough, Epistaxis | 8180 | 6314 | -22.8 | 1.17 | 1.35 | 15.4 |
| | 895 | Family Planning | 8166 | 6970 | -14.6 | 1.20 | 1.32 | 10.0 |
| | 780 | Vertigo, Dizziness, Headache | 8157 | 8321 | 2.0 | 1.21 | 1.39 | 14.9 |
| | 599 | Hematuria | 6544 | 7313 | 11.8 | 1.22 | 1.43 | 17.2 |
| | 727 | Tendonitis, Bunion | 5617 | 4540 | -19.2 | 1.24 | 1.36 | 9.7 |
| | 796 | Fatigue | 5552 | 5327 | -4.1 | 1.14 | 1.26 | 10.5 |
| | 709 | Mole, Other Disorders of Skin | 5471 | 5353 | -2.2 | 1.11 | 1.20 | 8.1 |
| | 847 | Strain, Sprain, Back Pain | 5165 | 4206 | -18.6 | 1.35 | 1.51 | 11.9 |
| | 272 | Hypercholesterolemia | 4941 | 4677 | -5.3 | 1.16 | 1.21 | 4.3 |

*The most common description associated with each diagnostic code is provided above; a full list of all diagnoses associated with each code is provided in the supplementary appendix.

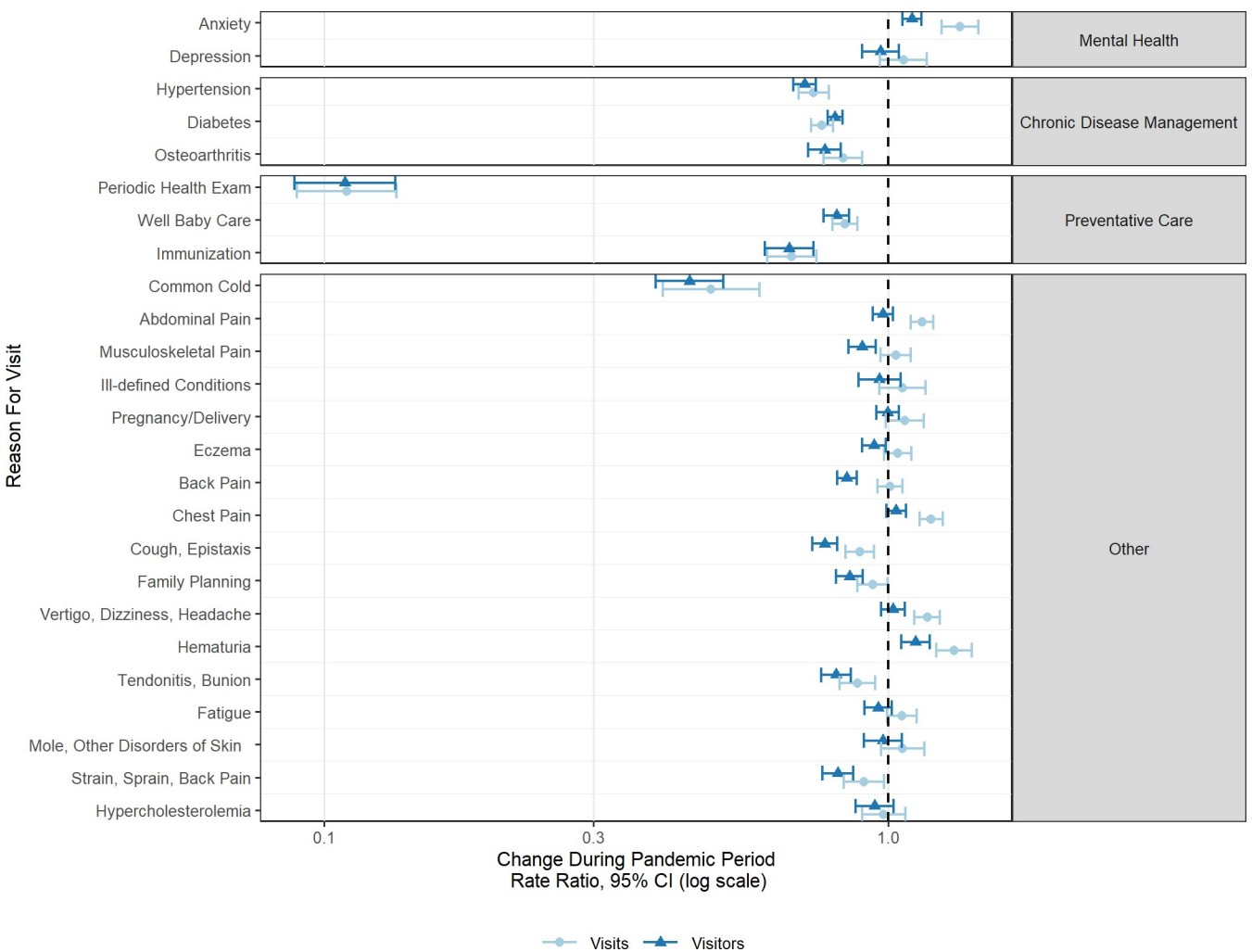

**Fig 2. Change in visit and visitor volume by reason for visit during the COVID-19 pandemic.** When the rate ratio for visits in larger than the rate ratio for visitors (i.e., interval is further to the right), this indicates that the intensity of visits per visitor increased during the pandemic. When the rate ratio for visits in smaller than the rate ratio for visitors (i.e., interval is further to the left), this indicates that the intensity of visits per visitor decreased during the pandemic.

was the only diagnostic code where the intensity of visits relative to the number of visitors decreased during the pandemic. The number of visits per visitor increased the most for mental health concerns with a 21.6% and 10.2% increase in the number of visits per visitor for anxiety and depression, respectively.

Fig 2 shows the extent to which the change in the number of visits associated with each diagnostic code is the result of a change in the frequency of distinct visitors and/or a change in the intensity of service use among visitors.

## Virtual care uptake by reason for visit

During the pandemic, virtual care became the dominant care format with virtual visits accounting for 77.5% of all primary care visits. Visits for immunizations, periodic health exams, well-baby care and pre-natal care were most likely to continue occurring in person, with less than half of these visits occurring virtually (Table 4). Visits for anxiety and depression were among the most common reasons for a virtual visit, with 90.6% of visits for these concerns during the pandemic occurring virtually.

**Table 4. Rate of virtual visits during the COVID-19 pandemic for the most common diagnoses.**

| Diagnostic code | Most common associated description* | Number of virtual visits | Percentage of visits involving virtual care |
|---|---|---|---|
| 272 | Hypercholesterolemia | 5142 | 91.2 |
| 300 | Anxiety | 55370 | 90.6 |
| 311 | Depression | 7678 | 90.4 |
| 460 | Common Cold | 9971 | 89.3 |
| 599 | Hematuria | 9064 | 86.5 |
| 786 | Cough, Epistaxis | 7296 | 85.7 |
| 796 | Fatigue | 5661 | 84.2 |
| 847 | Strain, Sprain, Back Pain | 5310 | 83.9 |
| 724 | Back Pain | 9810 | 83.6 |
| 781 | Musculoskeletal Pain | 17784 | 81.9 |
| 787 | Abdominal Pain | 19911 | 81.7 |
| 780 | Vertigo, Dizziness, Headache | 9400 | 81.4 |
| 799 | Ill-defined Conditions | 16589 | 80.7 |
| 785 | Chest Pain | 9454 | 79.1 |
| 691 | Eczema | 10691 | 77.3 |
| 715 | Osteoarthritis | 6540 | 76.5 |
| 401 | Hypertension | 22634 | 75.9 |
| 709 | Mole, Other Disorders of Skin | 4691 | 72.9 |
| 250 | Diabetes | 23531 | 70.8 |
| 727 | Tendonitis, Bunion | 4300 | 69.9 |
| 895 | Family Planning | 6208 | 67.4 |
| 896 | Immunization | 2708 | 46.3 |
| 650 | Pregnancy/Delivery | 7867 | 40.9 |
| 916 | Well Baby Care | 3716 | 22.6 |
| 917 | Periodic Health Exam | 770 | 22.1 |

*The most common description associated with each diagnostic code is provided above; a full list of all diagnoses associated with each code is provided in the supplementary appendix. One visit was counted per patient per date and less than 1% of visits involved the use of multiple diagnostic codes.

## Discussion

There were marked changes during the COVID-19 pandemic in the top reasons for visiting a family physician, when compared with before the pandemic, with substantial changes in the frequency of some types of visits more than others. Overall, visits for chronic disease and preventive care decreased, while visits for mental health concerns increased. For many conditions (e.g., pregnancy, skin disorders, various types of pain), there was no change in the number of patients accessing care but the intensity of visits per visitor increased. Taken together our findings highlight some important patterns of change in primary care visits that have implications for the delivery of primary care moving forward.

### Increased demand for mental health support

Anxiety and depression have been among the most common reasons for consulting a family physician in recent decades [14, 22], and the pandemic has increased the demand for these services even more. Before the pandemic, 7.6% of primary care visits in our sample were for anxiety or depression and this increased to 10.5% after the onset of the pandemic. This is consistent with concurrent reports from primary care physician surveys, with 61–76% reporting an increase in the number of patients with mental or emotional health needs [31, 32]. Our

findings suggest that both the number of individuals presenting in primary care with anxiety and the number of visits per visitor increased during the pandemic. The ability to offer virtual visits over the telephone or video may have allowed physicians to conduct follow-up appointments more regularly with patients presenting with symptoms of anxiety, resulting in a greater number of visits per patient on average. Results from representative samples of the general population suggest that rates of anxiety and depression have increased since the onset of the COVID-19 pandemic [33–35], but this has not necessarily resulted in an increase in anxiety/depression diagnoses made by primary care physicians [36, 37]. More research is needed to understand how family physicians are responding to the increase in mental health concerns and effective approaches for addressing the needs of distressed patients.

## Reduced preventive care

In Ontario, periodic health visits that focus on preventive care represent 4–8% of services provided by family physicians [38]. We found that periodic health visits ceased almost entirely with the onset of the COVID-19 pandemic, which is consistent with other reports of physicians limiting and patients avoiding seeking preventive care [39, 40]. There are concerns that the pandemic has led to a reduction in preventive services that are often delivered as part of a periodic health visit, including cancer screening [41–43], vaccinations [44, 45], and screening for chronic diseases [46], and that this will result in delayed diagnosis and worse disease outcomes. On the other hand, past research has found little evidence that general health checks, including periodic health visits, reduce morbidity and mortality for patients [47, 48]. The pandemic has created the conditions for a "natural experiment" that may help elucidate the value of visits focused on preventive care. Within the UTOPIAN database, there is now a cohort of patients who can be followed over time to identify potential health consequences that may result because of missed/delayed periodic health exam visits.

Adding to these concerns, we found that primary care visits where the main reason for the visit was immunizations dropped by 32.6%. Disruptions to the provision of routine childhood immunizations and decrease in adequate coverage for vaccine-preventable diseases due to the pandemic have been reported worldwide [49–52]. The volume of well-baby visits (during which childhood immunizations are usually provided) was 16.2% lower during the pandemic in our studied cohort. More research is needed to determine if this decrease has translated into missed or delayed routine immunizations in our patient population.

## Reduced care for chronic disease

We found that during the pandemic, diabetes, hypertension, and osteoarthritis continued to be among the most common reasons for consulting a family physician, but that the frequency of visits and the number of unique individuals accessing care for these chronic conditions decreased. Many primary care physicians in Canada and the United States reported limiting chronic care on surveys conducted at the start of the pandemic [39, 53], and continue to do so almost a year later [54]. The reduced care being provided for chronic conditions may result in poor disease control among those who already have the condition (e.g., higher Hemoglobin A1C, higher blood pressure) as well as less disease prevention for those at risk. The format of care delivery is also important to consider, given that 75.9% of visits for hypertension and 70.8% of visits for diabetes in the current study occurred virtually. A study of primary care visits in the United States found that blood pressure and cholesterol levels were less likely to be assessed and new treatments for hypertension were less likely to be initiated in virtual visits relative to office visits [55]. Efforts have been made to adapt current guidelines to support the management of chronic disease in primary care during COVID-19, including the use of virtual

care for patients with type 2 diabetes [56] and hypertension [57]. However, more research is needed to provide evidence of the effectiveness of these new approaches to chronic disease management.

## Limitations

Although the top 25 diagnostic billing codes assessed in the current study covered the majority of the primary care visits that occurred and these codes provide information about the primary reason for a visit, they do not necessarily capture all the issues addressed in a visit. Our findings should be interpreted with respect to changes in the clinician reported reason for visit, rather than the number of patients with a specific diagnosis. Data were drawn from a relatively large sample of physicians practicing in a high COVID region. Nevertheless, the findings are not necessarily representative of the experience across Ontario or in other health systems, especially those with lower COVID-19 case numbers. Our study compared primary care visits in only 2 time periods (pre-pandemic vs. during pandemic). We do not know to what extent the changes we observed included longer term trends in health services. However, the large effect sizes we observed are not consistent with past longitudinal research [58, 59], where a change in health service use of more than 30% occurred across 5 years rather than only 1 year. Although we were able to describe how the reasons for consulting a family physician changed after the onset of the COVID-19 pandemic, we do not know to what degree these changes are the result of changes in disease prevalence in the population (e.g., reduced rates of respiratory illness), reluctance among patients to seek healthcare services during the pandemic, and/or changes in provider availability or accessibility (e.g., providers unable to do in-person visits due to lack of personal protective equipment). It is likely that multiple factors are contributing to the changes we observed.

## Conclusion

The COVID-19 pandemic has changed how and why patients are accessing primary care services. Not only have more primary care visits occurred via telephone and video but the reasons associated with those visits have also changed. Visits for mental health concerns have become more frequent, accounting for a larger proportion of the total visit volume in primary care. Prenatal and well-baby were most likely to occur in-person, rather than virtually. Some of the largest decreases seen for primary care visits were for preventive care and common chronic conditions, with fewer individuals accessing care. Continued attention should be paid to the implications of these changes in how family medicine is being practiced and potential impacts on health outcomes for patients.

## Supporting information

**S1 Appendix. Inferred diagnoses for missing diagnostic codes.**
(DOCX)

**S2 Appendix. Data quality criteria.**
(DOCX)

**S3 Appendix. Eligible service codes.**
(DOCX)

**S4 Appendix. Full diagnostic descriptions.**
(DOCX)

## Acknowledgments

Dr's K Tu, D Butt, B O'Neill, N Crampton, C Ji receive Research Scholar Awards from the Department of Family and Community Medicine and/or the Rathlyn Foundation Primary Care EMR Research and Discovery Fund at the University of Toronto.

## Author Contributions

**Conceptualization:** Ellen Stephenson, Debra A. Butt, Jessica Gronsbell, Catherine Ji, Braden O'Neill, Noah Crampton, Karen Tu.

**Data curation:** Ellen Stephenson.

**Formal analysis:** Ellen Stephenson.

**Funding acquisition:** Debra A. Butt, Catherine Ji, Braden O'Neill, Noah Crampton, Karen Tu.

**Methodology:** Ellen Stephenson, Jessica Gronsbell.

**Supervision:** Karen Tu.

**Visualization:** Ellen Stephenson.

**Writing – original draft:** Ellen Stephenson.

**Writing – review & editing:** Ellen Stephenson, Debra A. Butt, Jessica Gronsbell, Catherine Ji, Braden O'Neill, Noah Crampton, Karen Tu.

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
