## [Decision Letter · Decision Letter 0]

14 Jun 2021

PONE-D-21-12572

Changes in the top 25 reasons for primary care visits during the COVID-19 pandemic in a high-COVID region of Canada

PLOS ONE

Dear Dr. Stephenson,

Thank you for submitting your manuscript to PLOS ONE. After careful consideration, we feel that it has merit but does not fully meet PLOS ONE’s publication criteria as it currently stands. Therefore, we invite you to submit a revised version of the manuscript that addresses the points raised during the review process.

We look forward to receiving your revised manuscript.

Kind regards,

Christine Leong, Pharm. D.

Academic Editor

PLOS ONE

Journal Requirements:

1. Please ensure that your manuscript meets PLOS ONE's style requirements, including those for file naming. The PLOS ONE style templates can be found athttps://journals.plos.org/plosone/s/file?id=wjVg/PLOSOne_formatting_sample_main_body.pdf and https://journals.plos.org/plosone/s/file?id=ba62/PLOSOne_formatting_sample_title_authors_affiliations.pdf

Additional Editor Comments (if provided):

Thank you for your submission of a very interesting paper. It is interesting to observe the extent of change in reasons for primary care visits pre and during pandemic. Very well written. A few comments raised by the reviewer and below could be addressed:

Title page

- Please identify affiliation 7 for Karen Tu

Data Source

- Please provide additional details about the COVID-19 rates in the population. It was mentioned the area included were in areas of high rates, please consider defining what is high for readers to assess generalizability. Also consider commenting on public health measures that were implemented during the study period

Limitations

- Edit sentence in line 252 “ Data were from drawn from“

- Recommend addressing that only two time periods were measured. You mentioned that it was difficult to tease out degree of prevalence change. I would further elaborate that it is uncertain of the trend in seeing the physician for anxiety has increased over time (ie. How do we know the rate of change between 2019 and 2020 was not expected. Analysis of additional pre pandemic years May have been beneficial

Reviewers' comments:

Reviewer's Responses to Questions

**Comments to the Author**

1. Is the manuscript technically sound, and do the data support the conclusions?

Reviewer #1: Yes

2. Has the statistical analysis been performed appropriately and rigorously? 

Reviewer #1: I Don't Know

3. Have the authors made all data underlying the findings in their manuscript fully available?

Reviewer #1: Yes

4. Is the manuscript presented in an intelligible fashion and written in standard English?

Reviewer #1: Yes

5. Review Comments to the Author

Reviewer #1: Statistics:

I selected "I don't know" in the statistics section because of Table 4 (Effect of COVID on the number of visits etc.). I am trying to rationalize why an IRR was calculated as this usually is done when there's a set exposure (exposed vs not exposed) and a certain health outcome over a time period. Practically speaking, if this paper is being geared towards family physicians, interpreting this table by sifting through confidence intervals to understand the text in the results/discussion is tedious and I'm not sure was actually necessary. Just a suggestion, but could the results be presented as "hard numbers" - for example, actual # of visits for Diabetes (% of total visits in brackets) and then percentage change pre/post COVID? That would be much easier to understand. Perhaps placing an * by the statistically significant results? I feel that the actual visit numbers and percentages should be presented so the reader is able to interpret outcomes themselves. This data is very appropriately presented in Table 2, and perhaps the change in % could be added in another column to the right?

I have attached a 2 recent papers looking at very similar outcomes. 1) In an Ontario population published in CMAJ Feb 2021 - Looking at their statistical analysis, they used 2-sample z tests, which I think makes more sense if this is a 2-group, independent sample with a large sample size (>50). 2) A second paper from the US that used simple descriptive statistics.

I'd strongly recommend rethinking Table 4 and revisiting the statistical analysis section for appropriateness of IRR after reviewing other similar papers.

Abstract:

If your group decides to present Table 4 differently (or take it out completely), I'd then suggest to edit the abstract as there are IRR values with 95% CI's. I was expecting in the abstract to potentially see the "top 5" visit reasons pre- and post- COVID summarized as that was your primary outcome. I liked how the anxiety outcome was presented in the abstract and I'd suggest presenting other outcomes the same way in the results section. I'd also suggest defining your secondary outcome as percentage of virtual care visits (clarifying this in the paragraph before the methods (lines 71-73) and including that as well in your abstract.

Table 1:

In Table 1 (Physician and Patient characteristics), would it be possible to include what the income quintiles are (in $ values)?

Discussion:

In the Discussion, in the first paragraph, suggest highlighting visit reasons that were NOT different pre- and post- COVID.

I think this paper definitely adds to the body of knowledge, especially since it defines which visit types changes substantially with COVID and advocates for more mental health supports which are desperately needed. It was well-written.

6. PLOS authors have the option to publish the peer review history of their article (what does this mean?). If published, this will include your full peer review and any attached files.

Reviewer #1: **Yes: **Grace Frankel

---

## [Author Response · Author response to Decision Letter 0]

19 Jul 2021

Response to Editor and Reviewer Comments

Editor Comments:

Data Source

- Please provide additional details about the COVID-19 rates in the population. It was mentioned the area included were in areas of high rates, please consider defining what is high for readers to assess generalizability. Also consider commenting on public health measures that were implemented during the study period

Authors’ response: We have added the following text to the Method to elaborate on the COVID-19 case numbers and public health interventions in the Greater Toronto Area and across Ontario throughout 2020. 

“The number of confirmed cases of COVID-19 per capita was more than 1.5 times higher for public health units in Peel (2.5 per million) and Toronto (1.9 per million) relative to the provincial average (1.2 per million) as of December 31, 2020 [24]. Public health measures implemented to control the spread of COVID-19 included closures of schools and non-essential businesses, travel restrictions, instructions to only leave home for essential purposes such as accessing medical services, mandatory mask wearing in indoor public spaces, and limiting the size of social gatherings.”

Limitations

- Edit sentence in line 252 “ Data were from drawn from“

- Recommend addressing that only two time periods were measured. You mentioned that it was difficult to tease out degree of prevalence change. I would further elaborate that it is uncertain of the trend in seeing the physician for anxiety has increased over time (ie. How do we know the rate of change between 2019 and 2020 was not expected. Analysis of additional pre pandemic years may have been beneficial

Authors’ response: We have revised the limitations section as recommended. We acknowledge that longer term time trends (i.e., expected increases or decreases for specific types of visits) could not be accounted for using our current design. However, large changes across a 1-year period are not consistent with past longitudinal research describing trends in health service use. We chose not to include additional pre-pandemic years because this would introduce additional limitations with respect to EMR data quality, completeness, and availability across the patient population. The following text was added to the Limitations section:

Our study compared primary care visits in only 2 time periods (pre-pandemic vs. during pandemic). We do not know to what extent the changes we observed included longer term trends in health services. However, the large effect sizes we observed are not consistent with past longitudinal research [58,59], where a change in health service use of more than 30% occurred across 5 years rather than only 1 year.

Reviewer #1 Comments:

Statistics:

I selected "I don't know" in the statistics section because of Table 4 (Effect of COVID on the number of visits etc.). I am trying to rationalize why an IRR was calculated as this usually is done when there's a set exposure (exposed vs not exposed) and a certain health outcome over a time period. Practically speaking, if this paper is being geared towards family physicians, interpreting this table by sifting through confidence intervals to understand the text in the results/discussion is tedious and I'm not sure was actually necessary. Just a suggestion, but could the results be presented as "hard numbers" - for example, actual # of visits for Diabetes (% of total visits in brackets) and then percentage change pre/post COVID? That would be much easier to understand. Perhaps placing an * by the statistically significant results? I feel that the actual visit numbers and percentages should be presented so the reader is able to interpret outcomes themselves. This data is very appropriately presented in Table 2, and perhaps the change in % could be added in another column to the right?

I have attached a 2 recent papers looking at very similar outcomes. 1) In an Ontario population published in CMAJ Feb 2021 - Looking at their statistical analysis, they used 2-sample z tests, which I think makes more sense if this is a 2-group, independent sample with a large sample size (>50). 2) A second paper from the US that used simple descriptive statistics.

I'd strongly recommend rethinking Table 4 and revisiting the statistical analysis section for appropriateness of IRR after reviewing other similar papers.

Authors’ Response: The reviewer has raised a good point about how difficult it may be for many readers in our target audience (family physicians) to easily understand the statistics originally reported in Table 4. We have revised our presentation of the results in the text and in the tables to clarify the value added by each outcome measure, in particular the importance of looking at the number of unique individuals visiting at least once before and during COVID by reason for visit. 

We have added columns with percentage change year over year to Table 2 as recommended, and report percentage change in the number of visitors and in the number of visits per visitor in Table 3. We now present the IRRs and 95% confidence intervals originally reported in a Table 4 as a forest plot (Figure 2). We hope that this will make it easier for readers to quickly see which types of visits increased/decreased the most during the pandemic. 

A key difference between our statistical analysis and those used in the previous studies mentioned, is that we do not have 2 independent samples over time. Our data come from a fixed cohort of physicians for whom the number of visits and patients seen pre-COVID was compared to the number of visits and patients seen for the same dates after the onset of the pandemic. Thus, our analytic approach was chosen to account for the correlated nature of these paired observations. We have revised the Statistical Analysis section of the manuscript to make the justification for our approach clearer. 

Abstract:

If your group decides to present Table 4 differently (or take it out completely), I'd then suggest to edit the abstract as there are IRR values with 95% CI's. I was expecting in the abstract to potentially see the "top 5" visit reasons pre- and post- COVID summarized as that was your primary outcome. I liked how the anxiety outcome was presented in the abstract and I'd suggest presenting other outcomes the same way in the results section. I'd also suggest defining your secondary outcome as percentage of virtual care visits (clarifying this in the paragraph before the methods (lines 71-73) and including that as well in your abstract.

Author’s response: We have removed the IRR values and 95% CIs from results section of the abstract. We report effect sizes as percentages instead. We have also revised the abstract to emphasize which types of visits were most common (i.e., “top 5”) and how the frequency of these visits changed during the pandemic. We now mention the proportion of visits involving virtual care as a secondary outcome in the Methods section of the Abstract. 

Table 1:

In Table 1 (Physician and Patient characteristics), would it be possible to include what the income quintiles are (in $ values)?

Authors’ Response: This measure of neighbourhood income quintile does not map to specific income ranges in dollar values. Rather the income quintiles are indexed to the income range for each Census Metropolitan Area. This means that the same dollar amount, e.g., $80,000, could belong to a different income quintile in a major urban centre than it does for a person living in a smaller town. An advantage of this approach is that the quintiles have similar meanings across regions with the country even though there may be differences in the cost of living in each region. 

Discussion:

In the Discussion, in the first paragraph, suggest highlighting visit reasons that were NOT different pre- and post- COVID.

Authors’ Response: We have revised the first paragraph of the discussion section to mention both what has changed during the pandemic and what has stayed the same. 

Additional changes:

While conducting additional analyses to revise the results section, we discovered that EMR data from some physicians was not correctly and/or completely extract to the UTOPIAN database. As a result, data from 17 additional physicians were excluded from the sample used in the revised manuscript. This did not change any of our findings or conclusions.

---

## [Editor Report · Decision Letter 1]

28 Jul 2021

Changes in the top 25 reasons for primary care visits during the COVID-19 pandemic in a high-COVID region of Canada

PONE-D-21-12572R1

Dear Dr. Stephenson,

We’re pleased to inform you that your manuscript has been judged scientifically suitable for publication and will be formally accepted for publication once it meets all outstanding technical requirements.

Kind regards,

Christine Leong, Pharm. D.

Academic Editor

PLOS ONE
---

## [Editor Report · Acceptance letter]

2 Aug 2021

PONE-D-21-12572R1 

Changes in the top 25 reasons for primary care visits during the COVID-19 pandemic in a high-COVID region of Canada 

Dear Dr. Stephenson:

I'm pleased to inform you that your manuscript has been deemed suitable for publication in PLOS ONE. Congratulations! Your manuscript is now with our production department. 

Kind regards, 

on behalf of

Dr. Christine Leong 

Academic Editor

PLOS ONE